# Should Caregivers Also Be Included in Multicomponent Physical-Exercise-Based Interventions for People with a Neurocognitive Disorder? The Caregivers’ Perspective

**DOI:** 10.3390/geriatrics8050086

**Published:** 2023-09-01

**Authors:** Flávia Borges-Machado, Duarte Barros, Paula Silva, Pedro Marques, Joana Carvalho, Oscar Ribeiro

**Affiliations:** 1CIAFEL—Research Centre in Physical Activity, Health and Leisure, 4200-450 Porto, Portugal; du_barros20@hotmail.com (D.B.); psilva@fade.up.pt (P.S.); jcarvalho@fade.up.pt (J.C.); 2Faculty of Sports, University of Porto, 4200-450 Porto, Portugal; 3ITR—Laboratory for Integrative and Translational Research in Population Health, 4050-600 Porto, Portugal; 4Department of Education and Psychology, University of Aveiro, 3810-193 Aveiro, Portugal; psipedromarques@gmail.com (P.M.); oribeiro@ua.pt (O.R.); 5Centro de Investigação em Tecnologias e Saúde (CINTESIS), Departamento de Educação e Psicologia da Universidade de Aveiro, 3810-193 Aveiro, Portugal

**Keywords:** care partner, dementia, qualitative research, physical activity, exercise

## Abstract

Informal caregivers of people with neurocognitive disorders (NCDs) may play a decisive role in guaranteeing partners’ participation in community-based physical exercise interventions. However, little is still known about their perspective on being involved in such programs that are specifically designed for their partners. This study aimed to explore the views of caregivers of people with NCDs about taking part in a multicomponent physical exercise intervention with their partners and to explore the perceived impact of this program on those caregivers who enrolled in it. An exploratory qualitative study was conducted with 20 caregivers (67.5 ± 13.94 years; seven female) from the “Body & Brain” project. Ten took part in the physical exercise sessions (active-participating caregivers), and the others did not (social-participating caregivers). Data retrieved from semi-structured interviews were analyzed following a thematic analysis approach. Regardless of their participation level, all caregivers reported their inclusion to be important in enhancing their partners’ initiation and engagement in the sessions; also, they all identified personal gains. Active-participating caregivers reported exercise-related benefits on general health, enjoyment, and social connectedness. Social-participating caregivers considered this intervention an opportunity for respite and appreciated being involved only occasionally (i.e., occasional gatherings or telephone contacts). The findings support the inclusion of caregivers in physical exercise interventions designed for partners with NCDs, considering their decisive role in the partners’ adherence and engagement and due to the perceived gains. Future community-based interventions designed for people with NCDs should consider giving caregivers the opportunity to choose whether they want or not to be actively involved in the exercise sessions. Further studies with larger samples are needed to verify these results, comparing caregivers’ point of view at baseline and post-intervention.

## 1. Introduction

In 2019, according to the World Health Organization, 55.2 million people worldwide were diagnosed with a neurocognitive disorder (NCD) [1]. NCDs encompass a group of conditions that affect the brain and in which the primary clinical deficit is in cognitive function [2]. Presently, more than 21 million people in OECD countries are estimated to have a diagnosis of an NCD [3]. Informal caregivers, in 2019, spent over 89 billion hours providing unpaid care and support in daily activities [1]. Caregiving disproportionately affects women, particularly in low- and middle-income countries, who are responsible for nearly 70% of informal care worldwide [1,3]. Evidence suggests that the burden and stress associated with care responsibilities increase caregivers’ susceptibility to health complications and diseases [4] and that those experiencing higher levels of emotional or mental strain may be at higher risk for death [4,5]. 

Caregivers often do not have time to engage in interventions targeting health and well-being, such as physical exercise training [6,7,8], and therefore, they tend to be less physically active compared to non-caregivers [6,9] and present worse psychological health [10,11].

Previous controlled trials investigating the effectiveness of home-based physical exercise programs specifically targeting caregivers have consistently demonstrated that this type of intervention may positively impact their daily physical activity levels and quality of sleep and reduce the occurrence of weariness and subjective burden [9,12,13,14]. A recently published study by Madruga, Gozalo [15] highlights that a 9-month home-based individually tailored physical exercise intervention supervised by a personal trainer may even reduce caregivers’ risk of depression. However, individualized interventions require significant financial resources and personnel, which supports the relevance of group-based approaches conducted in community settings [15]. 

Despite the growing evidence attesting the physical exercise benefits on caregivers’ psychological health, very few interventions have evaluated the effectiveness of community-based physical exercise programs targeting caregivers of people with NCDs [14]. More research is also needed regarding the indirect benefits of interventions exclusively conducted with partners with NCDs (e.g., reduced burden of care due to improvements in the physical and cognitive functioning of people with NCDs) [6,16,17].

A growing body of literature acknowledges the shared beneficial effect of physical exercise interventions designed for caregiving dyads [6,18], in which caregivers and partners with NCDs are enrolled and participate together. Dyadic interventions may enhance social participation and overcome barriers to engagement in physical activity (e.g., having someone to look after the partners with NCDs while caregivers undertake the exercise sessions alone) [7,18]. As reported by Lamotte, Shah [6], exercise sessions may represent an opportunity for caregivers to foster a bonding experience and establish a mutual purpose with partners. Also, along with the general health-related benefits associated with regular exercise, such sessions may positively impact their caregiving capacities and burden. However, caregivers may consider that planned respite or access to other types of interventions (i.e., support groups) may be more advantageous to them than being enrolled in a physical exercise program [6,18]. 

When planning an exercise intervention specifically for people with NCDs, researchers should reflect on caregivers’ involvement, i.e., whether to offer a dyadic intervention, other simultaneously occurring activity, usual care, or the opportunity to take part in physical exercise sessions. Caregivers play a decisive role in guaranteeing people with NCDs participation, providing task assistance, transportation, and encouraging engagement [6,18,19,20], and for this reason, their involvement may be of utmost importance. 

Several studies analyzing adherence support strategies for physical exercise interventions targeting people with NCDs reinforce the centrality of the caregivers’ role in ensuring partners’ adherence, particularly as the clinical condition progresses [20,21,22]. In this regard, the American College of Sports Medicine recently highlighted a special consideration concerning the participation of caregivers: “there may be benefits of training and physical exercises with the caregivers to help provide support, motivation, and monitoring of safety” [23] (p. 386). However, so far, existing evidence does not clearly describe what type of involvement informal caregivers might have in physical exercise interventions designed to partner with NCDs (i.e., whether they have access to a simultaneously occurring activity vs. assuming a supervision role vs. taking part in physical exercise sessions). In fact, caregivers are often overlooked for participation and examined only secondarily [18]; little is known about their beliefs and perspectives on being involved in interventions specifically designed for people with NCDs [19]. It is therefore of particular relevance to better understand the caregivers’ preferences to inform future interventions’ design, dosage, and implementation [18].

The present qualitative study aims to explore the views of caregivers of people with NCDs about taking part in a community-based physical exercise intervention with their partners. Complementarily, it aims to explore the subjective impact of the physical exercise program on those caregivers who have exercised together with their partners with NCDs.

## 2. Materials and Methods

### 2.1. Design and Ethics

The “Body & Brain” project is a quasi-experimental controlled trial with a parallel design (ClinicalTrials.gov–ID: NCT04095962) that aims to investigate the effect of a multicomponent training (MT) exercise intervention on the physical and cognitive function of older adults diagnosed with major NCD [24,25,26], in which the present exploratory qualitative study takes part. Ethical approval was obtained from the Institutional Review Board of the Faculty of Sports of the University of Porto (Ref CEFADE22.2018). Written informed consent was obtained from legal representatives, significant person, or assigned main caregiver of individuals with NCD. 

### 2.2. Participants

This purposive sample comprised caregivers of individuals with NCD who were participating in a group-based physical exercise program that was conducted in several community settings in the Oporto Metropolitan Area (Portugal), specifically at a sports university setting, at a community center sports hall, and at a psychogeriatric unit devoted to dementia care located in a psychiatric hospital campus (i.e., with outpatients). Only caregivers of community-dwelling individuals were recruited for this qualitative study. The bi-weekly exercise sessions took about 60 min in nonconsecutive days. They were subdivided into three main parts: 10 min for warm-up (e.g., general activation, slow walking, mobility exercises), 35–45 min dedicated to specific training (e.g., balance/coordination, strength, and aerobic exercises), and 5 min for cool-down (e.g., mobility and stretching conditioning), as described in detail elsewhere [25].

Caregivers were invited to participate in this MT physical exercise program as class members, according to their availability and willingness. In other words, this program was designed for people with NCD and not as a dyadic intervention. 

Caregivers freely decided whether they wanted or not to participate in the physical exercise sessions before the 6-month intervention. Those who agreed to participate, from now on designated as “active-participating caregivers”, could perform the proposed exercises or support their partners during exercises if necessary. In contrast, caregivers who did not want to take part in the physical exercise sessions, from now on designated as “social-participating caregivers”, were invited to attend specific recreational activities (i.e., two festive gatherings) and received occasional telephone contacts (i.e., to clarify any doubts about their partners’ participation in the program). 

Twenty-two caregivers were contacted by telephone to participate in this qualitative study; only two caregivers were not included in the final sample—one refused to participate in the interviews, and the other was unavailable due to professional reasons. The semi-structured telephone interviews were conducted with both active- and social-participating caregivers and took place between April and May 2020. 

### 2.3. Data Collection

This study followed the 32-item checklist of the Consolidated Criteria for Reporting Qualitative Research (COREQ) [27]. The interview guide was developed based on previous similar studies [28,29,30,31]. The first author (F.B.M., gerontologist) contacted the caregivers of community-dwelling individuals to explain the study, request their participation, and ascertain their time availability. 

The semi-structured interviews were conducted by a trained qualitative research assistant (P.M., psychologist) who was not involved in the “Body & Brain” project, during a previously scheduled telephone call (in Portuguese). The interviews were recorded using an encrypted voice recorder and lasted around 40 min (between 20 and 90 min). Interviewees were reminded that anonymity was guaranteed, and they could refuse to answer any of the questions or withdraw at any moment. 

Questions aimed to explore caregivers’ point of view about taking part in a community-based physical exercise program designed for their partners diagnosed with NCD (Table 1). Caregivers who took part in the physical exercise sessions with their partners were also invited to freely talk about the impact this intervention had on their lives. Caregivers’ sociodemographic characteristics and caregiving situation were collected during baseline evaluations within the scope of the “Body & Brain” project [25].

### 2.4. Data Analysis

Each interview was transcribed verbatim and reviewed for accuracy by the first author (F.B.M.). Interviewees’ identifying information was removed to ensure anonymity. The software analysis program NVIVO (Version 12, QSR International, Southport, UK) was used to assist with data management and analysis. Analysis of interviews was based on Braun and Clarke’s Thematic Analysis guidelines [32], using an inductive approach.

All interviews were coded by the first author (F.B.M.). An initial coding frame was developed, using the principle of constant comparison, to include data-driven themes and patterns. To ensure coding consistency, independent parallel coding was carried out by a second researcher (D.B.) who analyzed 20% of the transcripts. Subsequently, the two researchers met to reflect on and discuss the coding and analysis to ensure credibility. Discussion of the analytic process and findings was shared with a third author (O.R.), who further informed theme refinement. Disagreements regarding the identification, description, analysis, or interpretation of data were resolved through discussion with two additional co-authors (P.S. and J.C.) and further analysis. Trustworthiness was attained through category refinement via discussions and triangulations between researchers to reduce the effect of bias [32,33]. Illustrative verbatim quotations were included to support the interpretation of the identified themes.

## 3. Results

From the 20 caregivers who were interviewed, 10 were active-participating caregivers (mean age 72.9 years; range 59–83 years; 2 female), whereas the other 10 were social-participating caregivers (mean age 62.0 years; range 37–86 years; 5 female) who cited work or restrictive health-related issues (e.g., unstable cardiovascular disease and osteoarticular condition in which exercise is medically contraindicated) as the main reasons for not attending the physical exercise sessions with their partners. Most caregivers provided care daily, for more than ten hours. Four social-participating caregivers shared the task of providing care to their partner with an NCD, and five combined the task of caring with a working professional life. Table 2 summarizes caregivers’ characteristics. 

People with NCDs (mean age 74 years; range 65–84 years) were mostly female (75%), married (80%), and diagnosed with major NCDs (75%) due to multiple etiologies (40%) or Alzheimer’s disease (33%). The care partners presented a mean Mini Mental State Examination (MMSE) score of 18.6 (6.24) points and were highly independent in daily activities (Barthel Index > 80). 

Caregivers were asked about their perspective regarding taking part in a community-based physical exercise intervention designed for their care partners. Data analysis identified two common themes in both the active-participating and social-participating caregivers revealing their inclusive perspective on taking part in the intervention with their partners: “determinant to the partners’ participation” and “for own benefit”.

### 3.1. Active- and Social-Participating Caregivers

#### 3.1.1. Determinant to the Partners’ Participation

Both groups identified the inclusion of the caregiver in the exercise sessions as crucial, considering that it would play a key role in the partners’ participation, making them feel safe, supported, and motivated by knowing their limitations and capacities and, therefore, guaranteeing their adherence and involvement. Caregivers also mentioned the possibility to provide comfort and tranquility to partners throughout the intervention, whenever/if needed. 

“In my opinion, I think that she (partner with NCD) gets more motivated if she’s with the caregiver. (…) I think that it is great to include the caregiver!”(male, 67, spouse, active-participating caregiver, BB04)

“I think it’s pleasant and it brings benefits to the person (partner with NCD) as he feels accompanied, right?! (…) I think the person feels more confident. (…) He (partner with NCD) feels more protected.”(female, 56, daughter, social-participating caregiver, BB29)

#### 3.1.2. For Own Benefit

Caregivers perceived this program as an opportunity to fulfill their own needs for physical activity and recognized the potential benefits, both in terms of physical and mental health. 

“I think that it is very good that the caregiver is accompanying, in this case, I am accompanying my wife. (…) What would I do when she was on the exercise? Wait for her? At least I’m there to accompany her, and it is also beneficial for me.”(male, 76, spouse, active-participating caregiver, BB16)

“I think it’s good because it’s a great benefit for caregivers, whether they have a lot of [health] problems, like me, or not. Actually, I think it’s great!”(male, 86, spouse, social-participating caregiver, BB17)

Specifically, data retrieved from the interviews with active-participating caregivers about the impact of this physical exercise program showed the presence of two main perceived effects: “personal enjoyment” and “social connectedness”.

### 3.2. Active-Participating Caregivers

#### 3.2.1. Personal Enjoyment

Caregivers who enrolled in exercise sessions with their partners pointed out several positive aspects, the physical and mental health benefits being the most representative. Although physical exercise prescription was focused on their partners, these caregivers were fully involved during sessions and performed every proposed activity. Active-participating caregivers also described how they felt good and satisfied during sessions, enjoying this time for themselves.

“I always felt good (at the end of the exercise sessions). When it was time to go home, I used to say: Is it time already?! Today time flew, it was fast! (…) I have never disliked it… never, because if I did or if I felt sad or something else, I wouldn’t go, right? (…) I went there for her (the partner with NCD) and then I took advantage myself!”(male, 72, spouse, active-participating caregiver, BB126)

“As for physical health, as I said, I feel more relaxed. There is no doubt that that hour was sacred. It is very good, and I think it makes us feel lighter.”(male, 70, spouse, active-participating caregiver, BB134)

Active-participating caregivers also highlighted the major importance this program had on their daily routines and distinguished how they felt motivated being part of such a program with their partners; some even mentioned how being actively enrolled in the sessions positively influenced them in providing care. Several participants also outlined that during sessions, they did not play the role of caregivers, and, for this reason, these moments represented an opportunity for joy and even a break from their regular caregiving activities at home.

“It [participating in the program] was not a burden, it was an hour that passed quickly, and we even got entertained. (…) we are not used to having an hour twice a week outside our (house/caregiving) environment.”(male, 83, spouse, active-participating caregiver, BB116)

#### 3.2.2. Social Connectedness

Being able to socialize, particularly with peers living in a similar situation, was also an important perceived advantage for many caregivers. Taking part in physical exercise sessions allowed these individuals to get out of their caregiving environment (home), meet new people, and be entertained. They described how it felt good to talk with others and how it also helped them to put their personal problems in perspective. Above all, they enjoyed the familiar environment and the social interaction between participants and professionals and felt relieved while exercising. 

“The conviviality, the conversation, saying this or that, it entertained us a bit (…) Because it’s something that was a little out of the everyday routine, you see? And since it is out of the routine, we went there and then we talked about this and that (…) We become more relieved. (…) I’ve always enjoyed meeting people.”(male, 72, spouse, active-participating caregiver, BB126)

“At least I get out of home and socialize with others! If I have a big problem, I realized others may have a bigger one! (…) There, we are all the same and everything is good for us… we become more fulfilled and aware of the reality.”(male, 75, spouse, active-participating caregiver, BB130)

Along with the exercise-related benefits on general health, enjoyment, and social connectedness reported by active-participating caregivers, two common themes also emerged among social-participating caregivers regarding their involvement in the physical exercise intervention: “occasional involvement” and “opportunity for respite”.

### 3.3. Social-Participating Caregivers 

#### 3.3.1. Occasional Involvement 

Caregivers reported their appreciation for being enrolled in the program only in sporadic recreational moments (e.g., scheduled meetings with researchers for evaluations or festive gatherings) or by occasional telephone contacts. These individuals, either full-time workers or retired, considered that this option was appropriate, and, more importantly, it best met their own needs. 

“I honestly don’t say that I wouldn’t participate, but not on a regular basis, I mean, not in every class. I think the caregivers are not present (during exercise sessions) but you include them a bit from time to time in these snack parties, even in a meeting or in a conversation, or even over the phone (…) I think that the caregiver’s involvement [in sporadic recreational moments] is very important.”(female, 51, daughter-in-law, social-participating caregiver, BB26)

#### 3.3.2. Opportunity for Respite

Social-participating caregivers identified this intervention as an opportunity for respite during sessions. In other words, although acknowledging the importance of taking part in the sessions, some of them decided not to attend the bi-weekly sessions because it represented a break in their caregiving tasks and/or an opportunity to have some free useful time to deal with personal issues. 

“I don’t want to participate (in the exercise sessions) because I already spend 24 h with him (partner with NCD). And I wanted to be a moment away to rest a little. He goes one way, I go to another, and when we get home, we’re together again. (…) And while he goes to the gym (exercise sessions), I stay alone (referring to the activities/tasks that she does while the partner is on the exercise sessions) and I can feel free. (…) otherwise, I always have this burden. It sets me free! Some days are not easy, you know?”(female, 71, spouse, social-participating caregiver, BB124)

## 4. Discussion

This qualitative study explored the caregivers’ perspectives about being enrolled in a physical exercise program for their partners diagnosed with an NCD. Additionally, the perceived impact of this intervention was explored among caregivers who exercised with their partners. In summary, our study findings suggest that caregivers confirm the importance of their inclusion in the physical exercise sessions, not only due to their decisive role in their partners’ participation but also due to the associated physical and psychosocial potential effects. Moreover, those who were active participants reported a beneficial impact on general health, enjoyment, and social connectedness. Notwithstanding, caregivers should be given the opportunity to choose whether they want or not to be involved in physical exercise programs designed for people with NCDs, acknowledging their own needs. These data are particularly important when planning and prescribing future community-based exercise interventions for individuals with NCDs, considering the benefits that both people with NCDs and caregivers can obtain from participating in this type of intervention. Future physical exercise interventions should consider caregivers as potential participants of physical exercise sessions.

As mentioned by several authors, individuals with NCDs living in their homes depend to a large extent on their caregivers’ support to be physically active, get prepared to leave the family environment, arrive at sessions, and engage in community-based interventions [20,22,34,35,36,37,38,39]. In this study, caregivers were invited to enroll in the MT physical exercise program and take part in exercise sessions along with their family members considering their importance in guaranteeing the partners’ participation. According to their willingness and availability, they freely decided whether they wanted to participate in the bi-weekly sessions or only to be occasionally involved in recreational activities. Regardless of their type and level of participation, most caregivers agreed with their inclusion, recognizing their role in enabling partners’ access and compliance to a non-pharmacological therapy, namely by guaranteeing practical aspects like transportation and assistance with exercises and also motivating and boosting their partners’ confidence.

In line with Kim, Ullrich-French [19], our results suggest that future physical exercise interventions targeting people with NCDs should enhance caregivers’ confidence to support partners’ participation and/or encourage caregivers to increase their own daily physical activity levels, and/or promote caregivers’ awareness on health-related exercise benefits. Therefore, we consider that the usage of these strategies, along with the possibility to enroll caregivers in the sessions, may positively influence partners’ exercise adherence.

Findings from our study reveal that both groups of caregivers recognized the potential exercise-related benefits on their health and perceived this intervention as an opportunity to fulfill their physical activity needs while taking care of their loved ones and, for these reasons, agreed on their inclusion. Active-participating caregivers who were retired questioned the interviewer about what they would be doing during the exercise sessions if they could not attend, which may indirectly reflect how caregiving tasks are an essential structure in their daily routines [40,41]. These findings might reveal the lack of activities specifically targeting and/or including caregivers, even though they have shown interest in engaging in new experiences. As stated by Alzheimer’s Disease International, dementia caregiving dyads still suffer from stigma and social isolation and struggle to access post-diagnostic support opportunities [42,43]. Particularly in Portugal, although a national strategy for dementia was defined in 2018 (Diário da República, 2ª Serie, n.º 116, 19 June), and a formal statute for caregivers has been approved through a law (Portuguese ministerial order n.º 256/2020, 28 October; Law n.º 100/2019), these legislations and policies have not been reflected yet in additional availability and/or utilization of support services (e.g., information or advice on legal rights, financial benefits, social security protection, training and education, and respite services) [1,3]. 

When analyzing the perceived impact this intervention had on those caregivers who actively and regularly engaged in the physical exercise activities with their partners, personal benefits and enjoyment for participation were emphasized as valuable outcomes. Several caregivers even reported that they initially enrolled in exercise sessions to support their partner, but, after a while, they enjoyed the sessions for themselves. It is worth noting that this physical exercise program was prescribed to people with NCDs, and, therefore, it not always met caregivers’ physical fitness needs, even though individual adaptations were made whenever necessary following the international guidelines for exercise prescription [23,25]. In general, caregivers executed the proposed activities, which may explain the perceived benefits in physical and mental domains. Corroborating findings of the present study, a recent systematic review showed that caregivers are more likely to experience improvements in psychosocial and physical health when co-participating in exercise sessions compared to exercising alone [18].

Caregivers who participated in the sessions also reported that the program was essential for their daily routines, allowing them to get out of the home environment twice a week and promoting their motivation to better provide care for their family members. Some caregivers described how they felt relaxed during exercise sessions because they did not play their conventional role as caregivers, which facilitated their own engagement. These statements may reinforce the relevance of this novel strategy, which allows caregivers to decide what suits them better. Interestingly, active-participating caregivers did not mention anything regarding the opportunity to establish a mutual purpose and a bonding relationship with partners throughout the physical exercise program, as usually mentioned in the literature [6]. However, it was noticeable how important social interaction was for them. Caregivers appreciated the opportunity to meet new people and felt motivated and comfortable joining a group with others with whom they could share their experiences and who were in a similar situation. As stated by Long, Di Lorito [44], this shared experience of NCDs is significant for participants, and, although this was not the purpose of our program, caregivers mentioned on several occasions the relationship established with others, which was maintained even outside the context of the program. This social connectedness factor is thought to be important to overcome caregivers’ social isolation, and for this reason, it should also be considered when planning future group-based physical exercise programs. As previously highlighted by Guerra, Mendes [45] in a qualitative study regarding the evaluation of a multidimensional program for Portuguese informal caregivers of individuals with major NCDs, the opportunity to share experiences with others is crucial to attenuate emotional and social isolation and normalize some feelings and thoughts. Another recently published study [46], also conducted in Portugal, showed that a 10-week community-based psychoeducational program for primary informal caregivers of people with NCDs improved individuals’ mental health, satisfaction with care, and lessened caregiver strain. These findings strengthen the need for community-based interventions conducted in groups to counteract the lack of support and services available for caregivers. 

Finally, social-participating caregivers appreciated being involved in the program only sporadically in recreational activities. A bi-weekly physical exercise program for six months was perceived to be demanding, particularly for full-time working caregivers. In fact, this type of commitment may represent a burden for those who still carry professional responsibilities [47,48]. Other caregivers also mentioned physical limitations or conditions in which exercise is medically contraindicated as reasons for not being involved in this program. As stated by Lamotte, Shah [6], individuals who are older and/or have severe comorbidities may not be able to provide support and follow an exercise regimen, which reinforces the importance of offering them the possibility to be involved by telephone or in-person sporadic recreational moments. In addition, Lamotte and colleagues (2017) highlighted an important topic related to the increased time dyads must spend together if caregivers’ inclusion is mandatory for partners’ participation, which may increase the already high levels of distress. In this regard, and according to our study findings, the planned respite was perceived as an advantage for some caregivers [18]. 

The authors want to acknowledge some limitations that hinder the generalizability of this study’s results. First, active-participating caregivers’ adherence to physical exercise sessions was not registered, which may have biased the results considering that caregivers with higher adherence rates may have perceived more benefits from participating. Second, the active-participating caregivers’ actual involvement throughout the exercise sessions (e.g., if they supported partners in performing exercises or completed exercises independently) was not registered. Third, social-participating caregivers’ adherence to the pre-scheduled recreational activities was not equally registered, as well as the total number of telephone contacts with each one of the caregivers. Fourth, due to COVID-19 restrictions on conducting in-person interviews, it was not possible to collect the perspective of people with NCDs about the inclusion of caregivers. Fifth, the differences in age, gender, professional situation, and hours of providing care must be taken into consideration when comparing both groups. Lastly, the authors recognize that the small sample size, the innovative approach within the Portuguese context, and the exploratory nature of this study limit the replication of our study findings.

## 5. Conclusions

Overall, our results suggest that caregivers agree with their inclusion in community-based physical exercise programs designed for their partners with an NCD. Notwithstanding, caregivers should be able to choose whether they want to be actively or occasionally involved in the physical exercise sessions, acknowledging every individual reason/need to participate or not in the program. In contrast to specifically designed dyadic interventions, this physical exercise program did not require caregivers’ assistance and supervising roles, can be adapted to each caregiving situation, and promotes caregivers’ social integration in a dementia-friendly environment. In this regard, this approach may be seen as an alternative to physical exercise programs only designed for people with NCDs or caregivers, dyadic interventions, and those interventions that offer a simultaneous intervention to caregivers (e.g., support groups). Considering that the presented results come from an exploratory study, future community-based interventions designed for people with NCDs should consider giving caregivers the opportunity to choose whether they want or not to be actively involved in the physical exercise sessions and scientifically report the reasons presented. Further studies are needed to guide future implications and recommendations on physical exercise for people with NCDs and caregivers by the competent authorities.

## Figures and Tables

**Table 1 geriatrics-08-00086-t001:** Interview guide.

Active-Participating Caregivers	Social-Participating Caregivers
As you know, there are other physical exercise programs similar this one that do not include caregivers as participants on exercise sessions. What do you think about this program, particularly in what concerns the inclusion of the caregivers as attendees to exercise sessions?	As you know, there are other physical exercise programs like this one that include caregivers as participants on exercise sessions. What do you think about this program about not including the caregivers as attendees to exercise sessions?
This program has started in September 2019, with two sessions per week for 1-h. What has this program brought to your life? Has this program influenced your tasks as caregiver or burden? I would like you to explain me about how do you feel before and after exercise sessions	

**Table 2 geriatrics-08-00086-t002:** Caregivers’ characteristics.

Characteristics	Caregivers (n = 20)	Active-Participating Caregivers (n = 10)	Social-Participating Caregivers (n = 10)
Age (years), mean (SD)	67.5 (13.94)	72.9 (7.06)	62.0 (17.16)
Age, range	37–86	59–83	37–86
Gender (female), n (%)	7 (35)	2	5
Years of formal education, mean (SD)	7.9 (3.11)	7.1 (3.00)	8.7 (3.16)
Education, n (%)			
Medium (4–6 years)	6 (30)	4	2
High (7–12 years)	13 (65)	6	7
Superior (>12 years)	1 (5)	0	1
Professional situation, n (%)			
Working	6 (30)	1	5
Retired	14 (70)	9	5
Caregiver relationship, n (%)			
Spouse/partner	14 (70)	9	5
Adult children	4 (20)	0	4
Other relative	2 (10)	1	1
Number of years as a caregiver, mean (SD)	4.3 (2.36)	4.4 (2.41)	4.1 (2.42)
Sharing the task of providing care, n (%)	4 (20)	0	4
Frequency of care, n (%)			
Continuously	17 (85)	10	7
Working days/weekends/rotationally	3 (15)	0	3
Providing care (hours/day), n (%)			
0–3 h	5 (25)	0	5
4–7 h	2 (10)	1	1
8–10 h	1 (5)	1	0
>10 h	12 (60)	8	4

## Data Availability

The data that support the findings of this study are available from the corresponding author, upon reasonable request. The quotes and questions in Table 1 were translated from the Portuguese language by a qualified translator. The original versions in Portuguese are available upon request by any interested researchers.

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
