# Peer review of "Should Caregivers Also Be Included in Multicomponent Physical-Exercise-Based Interventions for People with a Neurocognitive Disorder? The Caregivers’ Perspective"

_geriatrics, 2023, doi:10.3390/geriatrics8050086_

Round 1
Reviewer 1 Report (Previous Reviewer 1)
I believe the authors have effectively responded to the reviewers' comments in the revised manuscript, so I have no additional remarks to add.
nil
Author Response
The authors want to tank you for your comment.
Reviewer 2 Report (Previous Reviewer 2)
Review for Geriatrics, Manuscript ID geriatrics-geriatrics-2557192: Should Caregivers Also Be Included in Multicomponent Physical Exercise-Based Interventions for People with a Neurocognitive Disorder? The caregivers’ perspective.
This is a second-round review. The authors have removed all of the “BLINDED FOR REVIEW” statements and substituted relevant information. I agree that the overall quality of English language is good. My initial review was too harsh.
However, the authors still seem to be presenting the paper as if the participants were English-speaking and were given instructions in English. The authors replied “The quotes were translated by a qualified translator, with experience in academic editing / proofreading scientific articles. The authors are available to share the Portuguese version of the quotes if requested by any interested researcher,” but the readers also need to be informed. I don’t find this information anywhere in the manuscript; and I don’t find anywhere a statement that the language used was Portuguese and that the quotes shown are translations, not the actual words used.
The language issue is important because it concerns how well participants understood the questions. Normally, one would strive to keep the language simple in structure and vocabulary to accomplish this goal, unless all participants happen to be highly educated. Also, more clarity would enable another group of researchers to reproduce the authors’ work, but such a reproduction seems impossible the way the manuscript is currently written.
English participants might struggle with understanding the words shown in Table 1, due to the level of vocabulary used (e.g., “elucidate”), the complex sentence structures, and grammatical errors. Are we to assume that the Portuguese text was equally challenging? Examples:
1. “What do you think about this program, particularly in what refers the inclusion of the caregivers as attendees to exercise sessions?” –The word “refers” as used here appears to mean that “the inclusion of the caregivers as attendees” is referred to “exercise sessions,” which makes no sense.
2. “What has this program brought to your live?” –The word after “your” needs to be a noun, which “live” is not.
3. “I would like you to elucidate me about how do you fell before and after exercise sessions” – The object of “elucidate” should be the matter that is being clarified, not the person for whom the clarification is given. The word “fell” asks about falls that occurred before and after exercise sessions, which seems like a reasonable question, but I suspect that the authors were actually interested in how the participants felt.
Good quality, but will need corrections by English-language editors prior to publication.
Author Response
Thank you so much for your comments. The authors have tried to address all of your suggestions.
The authors performed changes in the Table 1, and added a new data availability statement. Thank you.
Round 2
Reviewer 2 Report (Previous Reviewer 2)
No further comments.
Good quality, though some English-language editing is needed.
This manuscript is a resubmission of an earlier submission. The following is a list of the peer review reports and author responses from that submission.
Round 1
Reviewer 1 Report
The title of the manuscript, "Should Caregivers Also Be Included in Physical Exercise-Based Interventions for People With a Neurocognitive Disorder?", effectively raises the question of caregiver inclusion in physical exercise interventions for individuals with a neurocognitive disorder. It succinctly captures the main topic of the study and the target population. However, the title would benefit from increased specificity by specifying the particular neurocognitive disorder being studied. Additionally, further clarification or specification of the type or nature of the physical exercise interventions would enhance the title's clarity. Lastly, it would be helpful to provide a hint about the study's perspective or objective, such as advocating for inclusion, evaluating benefits or challenges, or exploring caregiver involvement. Addressing these aspects would improve the title's specificity and informativeness, enabling readers to grasp the study's focus more precisely.
The abstract of the manuscript effectively outlines the study's aim to explore the perspectives of caregivers of people with neurocognitive disorders regarding their participation in a physical exercise intervention and the perceived impact of the program on caregivers themselves. The abstract provides relevant information about the study design, participant characteristics, and key findings, such as the importance of caregiver inclusion in enhancing partners' engagement and the reported benefits experienced by active-participating and social-participating caregivers. However, the abstract would benefit from including additional context about the significance of studying caregiver involvement in physical exercise interventions for individuals with neurocognitive disorders. It should also provide more details on the methodology employed, such as the interview process and data analysis methods. Furthermore, discussing the broader implications of the findings and providing a concise conclusion summarizing the main contributions of the study would enhance the abstract's overall impact and comprehensibility.
The introduction provides a comprehensive overview of the global prevalence of neurocognitive disorders (NCD) and the significant role of caregivers in providing unpaid care. It effectively highlights the burden and health risks faced by caregivers, emphasizing the potential benefits of physical exercise interventions. The introduction also acknowledges the existing evidence supporting home-based physical exercise programs for caregivers and the need for community-based interventions targeting caregivers of people with NCD. However, the introduction could benefit from providing more specific details about the gaps in the literature and the limitations of previous research. Additionally, it would be valuable to clearly state the research questions or objectives of the present qualitative study. Overall, the introduction sets a strong foundation for the importance of including caregivers in physical exercise interventions for individuals with NCD, but could be further strengthened with clearer research aims and more explicit identification of research gaps.
The methods section provides a clear and detailed description of the study design, participant selection, data collection, and analysis procedures. The use of a quasi-experimental controlled trial design and the inclusion of caregivers participating in a group-based physical exercise program in community settings add strength to the study. The ethical considerations and obtaining written informed consent from caregivers demonstrate a robust ethical framework. The use of a semi-structured interview guide based on previous studies and adherence to the Consolidated Criteria for Reporting Qualitative Research (COREQ) checklist enhance the rigor of the qualitative data collection. The data analysis process, including coding and thematic analysis, followed established guidelines and involved independent coding by a second researcher to ensure consistency. Triangulation and discussions among the research team members further strengthened the trustworthiness of the findings. However, it would be beneficial to provide more information about the training and qualifications of the qualitative research assistant and the second researcher involved in coding. Additionally, details regarding the sample size determination and the rationale for selecting a purposive sample could enhance the transparency of the methods section. Overall, the methods section demonstrates a sound approach to exploring caregivers' views and experiences within the context of the physical exercise intervention for individuals with NCD.
The results section provides a comprehensive overview of the characteristics of the interviewed caregivers and presents their perspectives on taking part in a community-based physical exercise intervention with their partners diagnosed with NCD. The inclusion of direct quotations from the caregivers adds depth and authenticity to the findings. The identification of common themes, such as the importance of the caregiver's presence for the partners' participation and the benefits experienced by the caregivers themselves, demonstrates the richness of the data. The differentiation between active-participating and social-participating caregivers adds nuance to the analysis. Additionally, the results highlight the personal benefits and enjoyment reported by active-participating caregivers, as well as the social connectedness they experienced through the program. The occasional involvement and the opportunity for respite identified by social-participating caregivers provide valuable insights into their perspectives. Overall, the results section effectively presents the caregivers' viewpoints and provides meaningful insights into their experiences within the physical exercise intervention. However, it would be beneficial to provide more information on the selection process and characteristics of the participants, as well as any potential limitations or biases in the study. Additionally, it would be valuable to include information on data saturation to ensure that a sufficient number of interviews were conducted to capture the full range of perspectives.
The discussion section provides a thorough analysis of the findings, highlighting the importance of caregivers' inclusion in physical exercise programs for individuals with NCD. The study findings support the notion that caregivers play a crucial role in facilitating their partners' participation and adherence to exercise interventions. The discussion effectively draws connections to previous literature and emphasizes the need to enhance caregiver confidence, promote their own physical activity levels, and raise awareness of exercise benefits. The recognition of the personal benefits and enjoyment experienced by active-participating caregivers, as well as the social connectedness they gained from the program, adds depth to the discussion. The study's findings also shed light on the need for respite and occasional involvement options for caregivers. The limitations of the study, such as the small sample size and the absence of perspectives from individuals with NCD, are acknowledged. Overall, the discussion provides valuable insights into the significance of caregiver inclusion in physical exercise interventions and offers important recommendations for future programs. However, it would be beneficial to discuss the potential implications of the findings in terms of policy and practice, as well as to provide suggestions for further research and potential avenues for intervention development. Additionally, more clarity could be provided regarding the specific adaptations made for caregivers' involvement in the exercise sessions and the potential challenges faced in implementing this approach.
Mention the conclusion section seperately.
Nil
Reviewer 2 Report
Review for Geriatrics, Manuscript ID geriatrics-2501479-peer-review-v1: Should Caregivers Also Be Included in Physical Exercise-Based Interventions for People With a Neurocognitive Disorder?
The authors explore qualitatively the views of caregivers of people with neurocognitive disorders (NCD) who took part in a physical exercise intervention with their partners, divided into groups of 10 caregivers who participated actively in exercising with their partners (active-participating caregivers) and 10 who participated only occasionally (social-participating caregivers). The paper is well-organized and treats a subject of considerable interest. However, before this paper can be properly evaluated, it needs more transparency concerning the population from which participants were selected, and the language used.
I suggest the authors remove all of the “BLINDED FOR REVIEW” statements and substitute the relevant information. Please note that the reviewers are provided with the names and affiliations of the authors so there is no point in hiding information in the paper from the reviewers, and doing so makes it harder to assess potential cultural influences that could affect the results of the study. It is conceivable that the same interventions performed with a different population would have different outcomes.
Perhaps most importantly, the language used is of interest. The authors seem to be presenting the paper as if the participants were English-speaking and were presented with instructions in English, but clearly this cannot be the case. These are obviously translations from another language (Portuguese?). The original language should at a minimum be available as supplementary information, so that readers can check for subtle mistranslations or loss of context in translation, if they wish.
The whole document needs to be proofread by a native English speaker. The word choices are a bit uncommon or awkward and sometimes border on being incoherent, although overall the quality of the English language is good enough to be easily understood. The proofreader also needs to be familiar with language commonly used in scientific papers. For example, the words reliability and validity have precise meanings that are commonly understood, but the words credibility and trustworthiness as used in section 2.4 Data analysis, do not.